# A Definition of a Heywood Case in Item Response Theory Based on Fisher Information

**DOI:** 10.3390/e26121096

**Published:** 2024-12-14

**Authors:** Jay Verkuilen, Peter J. Johnson

**Affiliations:** Ph.D. Program in Educational Psychology, CUNY Graduate Center, New York, NY 10016, USA; jverkuilen@gc.cuny.edu

**Keywords:** item response theory, Heywood cases, Fisher information

## Abstract

Heywood cases and other improper solutions occur frequently in latent variable models, e.g., factor analysis, item response theory, latent class analysis, multilevel models, or structural equation models, all of which are models with response variables taken from an exponential family. They have important consequences for scoring with the latent variable model and are indicative of issues in a model, such as poor identification or model misspecification. In the context of the 2PL and 3PL models in IRT, they are more frequently known as Guttman items and are identified by having a discrimination parameter that is deemed excessively large. Other IRT models, such as the newer asymmetric item response theory (AsymIRT) or polytomous IRT models often have parameters that are not easy to interpret directly, so scanning parameter estimates are not necessarily indicative of the presence of problematic values. The graphical examination of the IRF can be useful but is necessarily subjective and highly dependent on choices of graphical defaults. We propose using the derivatives of the IRF, item Fisher information functions, and our proposed Item Fraction of Total Information (IFTI) decomposition metric to bypass the parameters, allowing for the more concrete and consistent identification of Heywood cases. We illustrate the approach by using empirical examples by using AsymIRT and nominal response models.

## 1. Introduction

Item response theory (IRT) models are members of the generalized linear latent variable modeling framework [1], abbreviated as GLLVMs. They represent an adaptation of structural equation modeling (SEM) to include binary, ordinal, nominal, or more rarely, other kinds of data, as well as continuous or discrete latent variables. IRT can be viewed as an adaptation of the factor model using a link function, e.g., probit, logit, cloglog, etc., to deal with the nonlinear relationships induced by categorical data [2,3,4]. These models have proven to be essential for the analysis of data from the educational, psychological, and health sciences, particularly as more advanced data gathering methods have been used. For example, while classical test theory (CTT) methods are reasonable in situations where a small number of forms with pre-specified items are employed, IRT becomes essential when using advanced techniques such as planned missingness/matrix sampling designs or adaptive testing. IRT is also better than CTT at dealing with mixed item types, e.g., a test made up of thirty traditional multiple-choice items, six fill-in the blank items, and four multi-point constructed response items. While the most common IRT models have historically been unidimensional, recent developments in IRT have enabled the use of multidimensional (MIRT) models to handle issues such as reading passage or item type effects that induce local dependence, among other issues. As such, there has been a convergence of the IRT and factor analytic traditions. In addition, IRT models have been broadened to handle features of data that have theoretical importance, such as the asymmetry of the item response function incorporated to cope with features of item complexity [5].

*Improper solutions* represent a feature of the broader class of latent variable models, i.e., a situation where one or more estimated model parameters contradict the inherent meaning of the parameters. Heywood [6] first noted this issue in the context of factor analysis, where in modern terms, estimated error variances can become 0 or even negative, but similar issues occur in all special cases of GLLVMs: zero variances in multilevel models, boundary conditional probabilities in latent class analyses, and so on. In IRT, improper solutions are often referred to as *Guttman items*, a term we feel is perhaps too narrow. For simplicity, to relate to the broader factor analytic literature and to honor the historical precedent, we use the term *Heywood cases* to refer to these various forms of improper solutions henceforth.

Based on experience with the linear factor model, a Heywood case is a strong signal that the model is misspecified, empirically unidentified, or both (e.g., [7,8]). Common causes include the following:Over-factoring, which will typically lead the factor correlation matrix to be non-positive definite or the loading matrix to be rank-deficient;Failing to account for cross-loadings, which leads to unpredictable effects;Including locally dependent items that are more strongly correlated than is consistent with the factor model, i.e., doublets;Insufficient sample size for the model to be fit;Sampling variability.

The best way to deal with a Heywood case is to avoid it entirely by careful design, e.g., by not including or rewording items that are strongly correlated due to excessive similarity of wording, which would thus lead to a doublet, and gathering datasets of sufficient size. For the analysis of item parameters, Heywood cases are not hugely problematic and can be handled in a number of ways, e.g., bootstrapping [9]. The scoring of latent variables presents a much bigger challenge, as we discuss below, and it can be quite difficult to deal with these issues, particularly if the number of observations and/or items is small relative to the complexity of the model [10]. In this case, it is not at all uncommon for a game of whack-a-mole to emerge, wherein alterations to the model to address one problem unmask a different one somewhere else.

In relatively simple UIRT models, e.g., the two-parameter logistic (2PL), the detection of Heywood cases is not especially difficult; the examination of the estimated item parameters and graphing the item response function (IRF) usually suffice. Restrictive models such as the Rasch model are unlikely to exhibit a Heywood case at all, although the presence of items that would generate a Heywood case in more flexible models will simply render the Rasch model a poor fit. That said, it can become burdensome with a large number of items and, as discussed below, is dependent on parameterization. This ease of detection no longer holds for more complicated models, where the number and/or interpretation of parameters often becomes difficult, if not impossible. In MIRT models (e.g., [11]), asymmetric IRT (AsymIRT) models (e.g., [12,13]), and semi- or nonparametric IRT models such as monotone polynomial (MONOPOLY) models (e.g., [14]), among others, Heywood case detection becomes much more convoluted due to the lack of easily interpretable parameters.

The goal of this paper is to propose a strategy for detecting Heywood cases in IRT models that is less subjective than the simple graphing of the IRF and less reliant on rules of thumb based on specific parameterizations than existing methods are. Ideally, given a predefined standard, an IRT program could check for Heywood cases automatically, rather than leaving things up to the researcher to complete by hand. The paper proceeds as follows: First, we discuss Heywood cases in the unidimensional linear factor model—IRT is, after all, an adaptation of this model. Second, we examine the nature of Heywood cases in IRT and propose measures that are better able to detect Heywood cases than usual practice. These are based on derivatives of IRT and, most notably, on the Fisher information function. Third, we consider some empirical examples showing these measures in practice. Finally, we discuss the proposed measures and indicate future directions.

## 2. Heywood Cases in IRT

We first introduce important IRT concepts through the unidimensional linear IRT and 2PL models and then consider properties of more general IRT models that are indicative of Heywood cases.

### 2.1. Linear Item Response Models

Mellenbergh [15] put the linear factor model, among others, into the IRT framework. To introduce our ideas, we focus on the unidimensional linear IRT model. Let n=1,…,N denote cases. Let $Y_{v}$, v=1,…,V be the items. It is traditional in IRT to denote latent variables by θ, and we follow suit in this and all subsequent discussion. We denote by $\eta_{v}$ the item easiness and by $\xi_{v}$ the item slope (i.e., loading or regression weight), with $U_{nv}$ being independent error terms: (1)$$Y_{nv}=\eta_{v}+\xi_{v}\Theta_{n}+U_{vn}.$$
This model imposes the following conditions: $U_{v}∼N\left(0,\psi_{v}^{2}\right)$, $\Theta ∼N(0,\phi^{2})$, and $\mathrm{cov}(U_{v},\Theta )=0$. Assuming local independence further implies that $\mathrm{cov}(U_{v},U_{v^{\prime }})=0$. Common identification constraints are set as ξ=1 or $\phi^{2}=1$. For simplicity and consistency with the broader IRT literature, we assume that $\phi^{2}=1$; thus, all ξ can be estimated.

Mellenbergh [15] shows that item information for the unidimensional linear factor model is simply
(2)$$I_{v}\left(\theta \right)=\frac{\xi_{v}^{2}}{\psi_{v}^{2}},$$
which, by using the Law of Total Variance, is
(3)$$\frac{\mathrm{var}(E\left(Y_{v}|\Theta =\theta \right))}{E(\mathrm{var}\left(Y_{v}|\Theta =\theta \right))},$$
i.e., the local squared signal-to-noise ratio in the neighborhood of Θ=θ, meaning that the information gives an idea of the power of the relationship between trait and item. Fisher information is based on the local curvature of the log-likelihood in the neighborhood of a parameter value ([3], §3.6.3), which, in this case, is a constant with respect to θ but varies for other models (see also [16], Chapter 7). Blyth [17] shows that this quantity is a local Rao divergence, a fact that holds more broadly for exponential families, e.g., Bernoulli, Poisson, multinomial, etc. One important subtlety that bears mentioning is the following: in IRT models, θ is not an item parameter and, depending on the formulation of the model, may not be a parameter at all.

As opposed to other indices, Fisher information is used because, for locally independent items, it is additive, i.e., $I_{T}\left(\theta \right)=\sum_{v=1}^{V}I_{Y_{v}}\left(\theta \right)$. Furthermore, for each individual item,
(4)$${\mathrm{rel}}_{Y_{v}}\left(\theta \right)=\frac{I_{Y_{v}}\left(\theta \right)}{1+I_{Y_{v}}\left(\theta \right)},$$
and
(5)$$\mathrm{rel}\left(\theta \right)=\frac{\sum_{v=1}^{V}I_{Y_{v}}\left(\theta \right)}{1+\sum_{v=1}^{V}I_{Y_{v}}\left(\theta \right)},$$
across items. This is the IRT analog of reliability at a particular point θ. Because I(θ) is constant for the linear IRT model, applying it gives the usual expression for factor analytic reliability.

While used for item analysis and scale construction, IRT models are also measurement models, i.e., they can, given observed indicators $\vec{y}$, generate a prediction for Θ [18]; see also Basilevsky [19], Brown and Croudace [20], Skrondal [1] (Chapter 7), and Skrondal and Rabe-Hesketh [21]. Scoring typically requires numerical optimization and/or integration using the posterior likelihood of $\Theta |\vec{Y}=\vec{y}$. In the case of the linear factor model, scoring can be written as simple formulas. There are many different kinds of scoring matrices, e.g., Bartlett, Thomson, and Moore-Penrose. The unidimensional linear model’s scoring equations are particularly simple, however, and provide insights into the effect of a Heywood case on scoring more broadly. The relevant matrices are given in McDonald or Basilevsky.

For the present discussion, we focus on Bartlett scores, but similar effects emerge with others. The resulting predicted Bartlett score for the $n^{\mathrm{th}}$ case can be written as a closed-form expression: (6)$${\hat{\theta }}_{n}^{\left(B\right)}=\underset{\mathrm{Total}\;\mathrm{Information}}{\underset{︸}{\frac{1}{\sum_{v=1}^{V}\frac{{\hat{\xi }}_{v}^{2}}{{\hat{\psi }}_{v}^{2}}}}}\times \underset{\mathrm{Weighted}\;\mathrm{Ave}.\;\mathrm{of}\;\mathrm{Obs}.}{\underset{︸}{\left(\sum_{v=1}^{V}\frac{{\hat{\xi }}_{v}y_{vn}}{{\hat{\psi }}_{v}^{2}}\right)}}.$$
The first term in the product is the inverse of the total information, which is analogous to *V* in an unweighted average. The second term is essentially an information-weighted average of the item responses. (Indeed, the quantity $\frac{\sqrt{\mathrm{var}(E\left(Y_{v}|\Theta =\theta \right))}}{E(\mathrm{var}\left(Y_{v}|\Theta =\theta \right))}$ is known as the optimal scoring weight. See, e.g., Hambleton and Swaminathan [22] (§6.4) or Lord [23] (§4.13).) It is very clear from this expression that very small values of $\psi_{v}^{2}$ are problematic, inducing items with small unique variances to dominate ${\hat{\theta }}_{v}$. The effective number of observations to predict $\theta_{n}$ would go to infinity even if the actual number of observations were small.

To simplify things, consider the standardized solution, in which $0\leq \psi_{v}^{2}=h_{v}^{2}\leq 1$ and $\xi_{v}^{2}=1-h_{v}^{2}$. This implies that
(7)$$I_{v}\left(\theta \right)=\frac{1-h_{v}^{2}}{h_{v}^{2}},$$
and, after a bit of algebra, the total information
(8)$$\sum_{v=1}^{V}I_{v}\left(\theta \right)=\sum_{v=1}^{V}\frac{1}{h_{v}^{2}}-V.$$
Consider two scenarios: $h_{1}^{2}$ = 0.1 vs. $h_{1}^{2}$ = 0.05, with other items $h_{2}^{2},\ldots ,h_{10}^{2}$ = 0.2. In the first case, the total information is 45, and the fraction of the total information accounted for by the first item is 20%. In the second case, the total information is 55, and the fraction accounted for by the first item is 35%. This has a strong effect on the degree to which $\hat{\theta }$ is determined by item 1.

#### Binary Item Response Models

The linear IRT model can be adapted for binary data, where instead of observing $Y_{v}$ directly, only the signs of $Y_{v}$ are observed. The most natural way to generate an IRT model for binary data from the linear IRT model involves the probit link, i.e., Φ(·), the CDF of the standard Gaussian, which is then inserted into the Bernoulli likelihood. To account for the loss of information compared with fully observed $Y_{v}$, additional identification constraints are needed. The most common is to force $\psi_{v}^{2}=1$ for each item. Bauer [24] shows that this choice implies that the other parameters are effectively defined relative to the variance of the noise distribution for each item; see also [25].

While the probit model is useful, it has become more common to use the logit link—the inverse of the standard logistic function—to handle binary data. Unlike the probit, there is a closed-form expression for the logit and inverse logit functions, and the model can be interpreted in terms of odds. A breakdown of the mathematics, as well as explanations for the strengths and utility of each link function, can be found in Bock and Gibbons [26]. The logit framework induces models such as the two-parameter logistic (2PL) and related models such as the three- and four-parameter (3PL and 4PL) models.

### 2.2. Definition of a Heywood Case in IRT More Broadly

In the IRT context, items that induce Heywood cases are frequently known as Guttman items due to the fact that they behave as if they were items in a Guttman scale [27], which have implausibly steep item response functions, occurring when $\beta_{v}\to \pm \infty$. These are often identified graphically, paired with knowledge of parameter values that are implausible in well-known models, such as the 2PL. Unfortunately, graphing only takes us so far; it is not particularly helpful, for example, when coping with multidimensional models or models that have unfamiliar parameterizations.

To gain a better understanding, consider the 2PL model for a binary item $Y_{v}$. Let θ be the latent variable, and let $\mathrm{Pr}\left(Y_{v}|\Theta \right)=\pi \left(\theta \right)$. Consider
(9)$$\pi \left(\theta \right)=\frac{e^{\eta +\xi \theta }}{1+e^{\eta +\xi \theta }},$$
the 2PL model written in slope-intercept form. In the 2PL, a Heywood case (also known as a Guttman item) happens when π(θ) becomes a step function at $\theta^{*}=-\eta /\xi$, which implies $|\partial_{\theta }\pi \left(\theta \right)|\to \infty$ at $\theta^{*}$ and 0 otherwise, which is equivalent to |ξ|→∞ at $\theta^{*}$ and 0 otherwise. This in turn implies I(θ)→∞ at $\theta^{*}$ and 0 otherwise. This is equivalent to logistic regression separation in the space of θ at $\theta^{*}$ [2] (§6.5). To define a Heywood case in IRT, we generalize from these observations. Let
(10)$$\{\theta^{*}\}=\left\{\theta :\partial_{\theta }I\left(\theta \right)=0⋂\partial_{\theta }^{2}I\left(\theta \right)\leq 0\right\},$$
i.e., the critical points of I(θ) that are local maxima. (The conditions for vector $\vec{\theta }$ are gradiant $\vec{\nabla }I(\vec{\theta })=\vec{0}$ and Hessian determinant $|H_{I(\vec{\theta })}|<0$. Although generalizable to multivariate situations, we focus on univariate θ.)

Of course, no situation will fit this perfectly, so we need approximate measures that are focused on the consequences of an approximate Heywood case. There are a number of possible measures:The first derivative of the IRF: $\partial_{\theta }\pi \left(\theta \right)$;Fisher information: I(θ);Item Fraction of Total Information (IFTI): shown below.

Each of these measures will be discussed more thoroughly below.

#### 2.2.1. First Derivative of IRF

The first derivative of π(θ) represents two important pieces of information. First, it is the marginal (marginal in the sense of econometrics, i.e., a derivative, not an integral [28]) effect of θ on $Y_{v}$, representing the change in $\mathrm{Pr}(Y_{v}|\Theta )$ for a infinitesimal change in θ. Second, because the IRF is a CDF, $\partial_{\theta }\pi \left(\theta \right)$ is proportional to the error density on the latent scale. For the 2PL model, $\partial_{\theta }\pi \left(\theta \right)$ is maximized at $\theta^{*}$, and it equals $\xi \pi \left(\theta^{*}\right)\left(1-\pi \left(\theta^{*}\right)\right)$, which simplifies to ξ/4. (We already mentioned optimal scoring weights (Hambleton and Swaminathan [22] (§§5.2, 6.4); Lord [23] (§§4.13, 5.13).) For binary models, they would be $w\left(\theta \right)=\frac{\partial_{\theta }\pi \left(\theta \right)}{\pi (\theta )(1-\pi (\theta ))},$ which represent another scaling that could be used; they are a rescaling of the first derivative by the conditional variance, which does not have so natural an interpretation as the derivative or information. These weights exhibit the issue with a Heywood case very clearly: $\partial_{\theta }\pi \left(\theta \right)$ diverging to infinity near $\theta^{*}$ and to 0 elsewhere implies that the weight is essentially infinite near $\theta^{*}$ and 0 elsewhere.

When using standard reasoning for logistic regression, when var(Θ)=1, as previously assumed, |ξ|>5 is suspect, and we tentatively adopt this as an indication of a likely Heywood case in analogy to a too large regression weight in logistic regression, wherein a unit change on the regression scale would lead to an extremely large change in predicted probability [29], e.g., a change from 0.01 to 0.5 or 0.5 to 0.99. This measure is useful but has some issues. First, it does not easily extend to polytomous IRT, particularly models such as Bock’s nominal response model [30] (NRM) or constrained versions of it that make up the divide-by-total family of IRT models, and second, the scale of derivatives requires great care when interpreted.

#### 2.2.2. Fisher Information

Fisher information, I(θ), is also a very useful measure that is similar to the first derivative but is standardized. In binary items,
(11)$$I\left(\theta \right)=\frac{{\left(\partial_{\theta }\pi \left(\theta \right)\right)}^{2}}{\pi (\theta )(1-\pi (\theta ))}.$$
This can be viewed as a squared marginal effect divided by the local standard deviation: (12)$$I\left(\theta \right)={\left(\frac{\partial_{\theta }\pi \left(\theta \right)}{\sqrt{\pi (\theta )(1-\pi (\theta ))}}\right)}^{2},$$
or, as stated previously, a ratio of between-to-within local variance. In the 2PL model, $I\left(\theta^{*}\right)=\xi^{2}\pi \left(\theta^{*}\right)\left(1-\pi \left(\theta^{*}\right)\right)$, and $\theta^{*}=-\eta /\xi$. Analytic expressions exist for models such as the 3PL, 4PL, or various asymmetric models, but they are unwieldy. More complicated formulas exist for polytomous models. Baker and Kim [31] have analytic formulas for common polytomous models; unfortunately they are very complex and frequently do not simplify in useful ways. Fortunately, it is not necessary to compute these analytically—numerical differentiation over a sufficiently fine grid provides excellent approximations for models with awkward expressions for Fisher information.

I(θ) is scale-dependent, and we recommend standardizing θ to have unit variance to remove the dependence; this is typical but not universal practice in IRT. However, the ordinate of I(θ) is more understandable than $\partial_{\theta }\pi \left(\theta \right)$ due to the fact that reliability at a point θ is
(13)$$\mathrm{rel}\left(\theta \right)=\frac{I(\theta )}{1+I(\theta )},$$
from which it can be ascertained that I(θ)=10 roughly corresponds to a reliability of 0.9, etc. High reliability of a single item relative to the other items of an assessment is indicative of a potentially problematic item. Additionally, var(Θ)=1, |ξ|>5, from the first derivative measure is equivalent to $I(\theta^{*})>6.25$, indicating a suspect item. Figure 1 shows the IRFs (Figure 1a) and IIFs (Figure 1b) for a theoretical typical, suspect, and problematic item. Note that though the IIFs are on a $\sqrt{I(\theta )}$ scale for ease of visual comparison, as the problematic item’s IIF peaks at I(θ)=25, the suspect item has $\sqrt{I(\theta )}=2.5$, which corresponds to I(θ)=6.25.

I(θ) is advantageous over $\partial_{\theta }\pi \left(\theta \right)$ because information units are typically easier to work with compared with the derivative scale. In addition, I(θ) is much more comprehensible for polytomous IRT models, for which there would be many category-specific derivatives. However, since I(θ) is itself not scale-free, care is still needed when interpreting Fisher information when θ is not standardized.

#### 2.2.3. Item Fraction of Total Information (IFTI)

If items are assumed to be locally independent, then Fisher information is additive. We propose to quantify the amount of “theta theft” inherent in a particular IRT model by the Item Fraction of Total Information (IFTI), where
(14)$$IFTI_{j}\left(\theta \right)=\frac{I_{j}\left(\theta \right)}{\sum_{k=1}^{J}I_{k}\left(\theta \right)}.$$
*IFTI* has a number of important advantages, most notably that it is unitless, unlike both $\partial_{\theta }\pi \left(\theta \right)$ and I(θ), which have units 1/θ and $1/\theta^{2}$, respectively. This is particularly helpful in light of the indeterminacy of the θ scale discussed in [23]. In the case of IFTI, the units cancel, so it becomes a pure proportion bounded between 0 and 1, for any regular model. See Verkuilen [32] for a discussion of the importance of regularity conditions on Fisher information in IRT. To interpret $IFTI_{j}\left(\theta \right)$, we tentatively propose a rule of thumb, in analogy to the benchmark used for leverage in regression diagnostics (e.g., [33]): $IFTI_{j}>2/J$ is troublesome, and $IFTI_{j}>4/J$ is clearly problematic. These situations represent items that are *too* responsible for predicting θ, much like high leverage points are for predicting *Y* in regression. An alternative scaling would be to multiply the IFTI by *J*, converting it into an effective number of items, rather than a percentage.

## 3. Empirical Examples

We provide two empirical examples showing how the information approach can be useful for identifying potential improper solutions in IRT. Our first example uses an AsymIRT model, the residual heteroscedasticity (RH) model [34]. Our second example uses Bock’s nominal response model [30] (NRM). Both models feature parameters that are difficult to interpret, and both are based on publicly available data; our R code is available as Appendix A. These examples were created by using R, version 4.4.0 [35].

### 3.1. Example 1: Binary Residual Heteroscedasticity Model

The first example we use is based on data taken from the Synthetic Aperture Personality Assessment (SAPA) intelligence items, which are discussed fully in Condon and Revelle [36]. The data are available in the hemp package for R [37] associated with Desjardins and Bulut [38], who analyze the data in a number of ways. We previously analyzed these data in Verkuilen and Johnson [39], where we noted that the item set exhibits problematic fits for the 4PL, monotone polynomial [14] (MONOPOLY), RH [34], and 3PL models but seems to be more reasonably fitted by the 3PL with an upper asymptote, Logistic Positive Exponent [12], and Gumbel-Reverse Gumbel models [39]. The RH, Logistic Positive Exponent, and Gumbel-Reverse Gumbel models are examples of asymmetric IRT models, which have been proposed as alternatives to models with asymptotes such as the 3PL, which have been criticized on a number of grounds including interpretability, identifiability, and scoring (e.g., [5,12,40,41]). For our illustration, we focus on the RH model, but the 4PL, 3PL, and MONOPOLY models all exhibit essentially the same behavior. By contrast, the Rasch and 2PL models exhibit a systematic lack of fit.

Let
(15)$$\Lambda \left(x\right)=\frac{e^{x}}{1+e^{x}},$$
i.e., the standard logistic CDF. The RH model’s item response function is given by
(16)$$\pi \left(\theta \right)=\Lambda \left(\frac{\eta +\xi \theta }{\sqrt{\frac{1+e^{-\delta \theta }}{2}}}\right).$$
This function reduces to the 2PL when the heteroscedasticity parameter, *δ*, is 0; the Rasch model would be a special case where all *δ* = 0 and all *ξ* are equal to a common value. The RH has been criticized for being difficult to interpret, much like heteroscedastic binary regression models more broadly [28]. Of course, similar criticisms have been leveled against the asymptote parameters in the 4PL family [41], and the MONOPOLY model lacks interpretable parameters [14]. More importantly, perhaps, Feuerstahler et al. [42] show that for regions of the parameter space, the IRF is non-monotonic. That said, out of the asymmetric IRT models, it appears to have good identification.

Before fitting parametric models, we examined the SAPA data more descriptively. To avoid prejudging the scale’s properties based on any specific parametric model, we used Mokken scaling, a type of nonparametric IRT [43] via the mokken package [44]. This analysis shows $H_{scale}$ = 0.36, which suggests moderate unidimensionality according to Mokken’s benchmarks. Only item MX55 has $H_{j}<0.3$, in this case 0.29. We also checked item monotonicity and noticed no notable patterns of pairwise violation. Item easinesses (${\hat{\mu }}_{j}$; proportion correct) and item $H_{j}$ are included in Table 1. Examining $H_{ij}$ suggests that the multidimensionality exhibited is likely due to the presence of four different types of items, especially the mental rotation items. Examinees find mental rotation very difficult in general [36], and the data may have a mixture of examinees who attempted the items and others who simply skipped them; without access to the unscored data, it is hard to tell. We used mirt, version 1.40 [45], to fit these data, using MML/EM algorithm, and we used nonparametric bootstrapping to estimate standard errors.

Our graphical output highlights the mental rotation items in bold and color. Figure 1, Figure 2 and Figure 3 show the IRFs, derivative plots, test information function (TIF), item information plots, and *IFTI* plots. While the IRFs (Figure 2a; mental rotation items colored) show some signs of asymmetry, in particular long left tails, as well as separation from the other items, it is not especially evident if they are problematic; when they are plotted separately, as is the default in mirt, it is difficult to see this effect. The derivative plots (Figure 2b) show the latent error densities, illustrating the asymmetry and substantial concentration of probability better than the IRFs do; the scale of derivatives is, unfortunately, difficult to interpret, as mentioned above, but the separation between types of items is evident. The TIF (Figure 3a) is clearly problematic, given the spike near θ≈1.5. The item information plots (Figure 3b) reflect the overall trend in the TIF, illustrating very clearly that two of the items (MR3 and MR4) are clearly Heywood cases and responsible for the spike in the TIF.

Figure 4a shows the *IFTI* plot. It is very evident that essentially all of the information for θ between 1 and 2 comes from the mental rotation items, hardly a desirable circumstance and likely indicative of multidimensionality. Figure 4b shows IFTI(θ) when the most problematic items, MR3 and MR4, are dropped. Items MR6 and MR8, while not clearly problematic in terms of item information, still explain almost all of the information between θ=1 and θ=2. Thus, even when the most problematic items are removed, others take their place. Item MR8 was already a problem in terms of the IFTI, but item MR6 also becomes a substantial problem. This suggests that because θ is not well identified in this range, with few items explaining a majority of the information, Heywood cases are not easily addressed simply by removing problematic items—especially on scales with low numbers of items, a situation where IRT is commonly used and desired. In addition, item MX55 has been bolded because it has a high IFTI for extreme values of θ induced by its low discrimination: other items have little information for high or low values of θ. Given that no examinees appear in this region of θ, it would be safe to ignore those high values.

#### Effect on Scoring

As we have stated previously, IRT is primarily a measurement model. Unlike the linear factor model, IRT scoring cannot be written as a simple closed-form expression; instead, it requires numerical optimization and/or integration [21]. To assess the effect on scoring in this model, we generated Expected a Posteriori (EAP) predicted scores ($\hat{\theta }$) and the associated standard errors ($\mathrm{SE}(\hat{\theta })$). These scores set an initially Gaussian prior for θ and then update the prior based on observed responses and estimated item parameters by using Bayes’ rule; this process requires numerical integration. We also examined other scoring methods but focus here on EAP for three reasons: First, EAP is the default scoring method in common software. Second, EAP provides useful shrinkage in a way that is analogous to Thomson scores in the linear model. Third, EAP is known to avoid some of the pathologies of the other scoring methods, such as maximum likelihood (ML), which can generate infinite $\hat{\theta }$ for perfect scores (i.e., all correct or incorrect). That said, the response patterns we highlight as being problematic are problematic with all scoring methods, and the same basic issues occur for all methods.

The box plots (Figure 5) we show are at each level of percent correct. Most scores appear to be quite reasonable, with some variation due to different response patterns in both $\hat{\theta }$ (Figure 5a) and $\mathrm{SE}(\hat{\theta })$ (Figure 5b). However, for higher percent correct, there is a substantial degree of variability that is unexpected for ordinary IRT scores. In addition, the standard errors fluctuate quite widely, with some being quite high, i.e., 0.5, and some being extremely low, i.e., 0.2. Examining the response patterns shows that this is primarily driven by whether the item or items the examinees got wrong included mental rotation. Having this much volatility induced in the predicted score and its associated standard error by one type of item seems highly undesirable.

Instead of just removing problematic items from the item pool, which can induce other items to become Heywood cases, among other problems, one potential solution to at least reduce the severity of Heywood cases is to impose regularizing priors on some of the parameters in the data. However, this brings a new set of challenges, like identifying a representative and informative prior and determining which parameters might need priors, among others. In the RH model, the asymmetry parameter seems to cause much of the problematic information observed. To address this, we added priors onto this parameter to assist in model estimation.

Figure 6a shows the IFTI for the SAPA items when Δ∼N(0,1), which is a relatively loose prior for the parameter. Note that while some of the mental rotation items have reduced IFTI, MR8 (red) still contributes substantially to the information in that range of θ. Additionally, when examining the EAP scoring (Figure 6b), there is not any noticeable reduction in the variability seen in the higher percent correct of predicted scores when compared with the RH model without priors. Thus, we applied a stricter prior onto the asymmetry, namely, Δ∼N(0,0.25). With this strict of a prior, Figure 7a shows a substantial reduction in the Heywood case severity of all four MR items, and Figure 7b shows a drastic decrease in variability in predicted scores for the higher percent correct.

However, the cost of stricter priors is a worse fit, at both the item and overall model level. In this case, the misfit was not egregious—the model RMSEA increased from 0.0333 in the model with no prior to 0.0342 in the model with the strict prior, with the loose prior falling in between. The item RMSEAs followed a similar inoffensive but noticeable increase, specifically in the four MR items, yet there was still no drastic misfit. However, observing the empirical groupings of θ on the item characteristic curves, the same systemic misfit of the 2PL when we initially fit these data begins to appear. This makes sense, however, as the 2PL exemplifies the strictest prior one can place on the asymmetry parameter in the RH model: forcing Δ=0. Unfortunately, while the strict Δ∼N(0,0.25) prior improves the scoring variability, the problems have not all disappeared; an even stricter prior is needed to better improve scoring for these data, but that comes at the cost of inducing misfit—see the discussion for more on this idea.

### 3.2. Example 2: Nominal Response Model

The nominal response model (NRM) [30,31,46] is a baseline multinomial logit IRT model for unordered categories, $Y_{v}=1,\ldots ,K$. For simplicity, we assume that the number of categories is the same for each item, although this is not, strictly speaking, necessary. The IRF, $\mathrm{Pr}(Y_{v}=k|\Theta )$, is
(17)$$\pi_{k}\left(\theta \right)=\frac{e^{\alpha_{k}+\gamma_{k}\theta }}{\sum_{k=1}^{K}e^{\alpha_{k}+\gamma_{k}\theta }}.$$
This equation requires identification constraints; typically sum-to-zero constraints, i.e., $\sum_{k=1}^{K}\alpha_{k}=0$, $\sum_{k=1}^{K}\gamma_{k}=0$, are used, but baseline category constraints with a different parameterization could also be used, e.g., as specified in mirt or the alternatives discussed in Thissen et al. [46].

Because of their sheer number, combined with the identification constraints, the parameters of the NRM are notoriously difficult to interpret directly. As such, it makes sense to consider category predicted probabilities ($\pi_{k}\left(\theta \right)$), category information ($I_{jk}\left(\theta \right)$), or total item information ($I_{j}\left(\theta \right)$). The information functions are based on the category information share
(18)$$I_{jk}\left(\theta \right)\pi_{k}\left(\theta \right)=\frac{{\left(\partial_{\theta }\pi_{k}\left(\theta \right)\right)}^{2}}{\pi_{k}\left(\theta \right)}-\partial_{\theta }^{2}\pi_{k}\left(\theta \right),$$
which, upon summing over *k*, gives
(19)$$I_{j}\left(\theta \right)=\sum_{k=1}^{K}\left(\frac{{\left(\partial_{\theta }\pi_{k}\left(\theta \right)\right)}^{2}}{\pi_{k}\left(\theta \right)}-\partial_{\theta }^{2}\pi_{k}\left(\theta \right)\right).$$
See Baker and Kim [31] (§9.4) for additional details.

For this example, we examine the SAT12 data, which are found in the mirt package but were originally distributed with TESTFACT [47]. These are 600 12th-grade examinees who took 32 five-option multiple-choice science items. There is a very modest number of missing data. Nonparametric IRT analysis of the binary scored data with Mokken scaling shows that the items seem to form a weak scale (H=0.20), suggesting that analysis with the NRM may be useful, for instance, to determine if there are mis-keyed items.

We fit these data by using mirt. Because the number of parameters is quite large (8 per item times 32 items, for a total of 256 parameters), we provide the parameter estimates as Appendix A, along with their bootstrap biases and standard errors, rather than reproducing them here. Figure 8a shows the item information curves, $I_{j}\left(\theta \right)$, and Figure 8b shows the IFTI. The TIF is unnecessary, as the overall information trend can be seen in the item information plots: a very high peak around θ=2. Item 11 is highlighted due to its problematic IFTI and very high item information value, seeming to cause the spike in information.

Figure 9a shows the values for the IFTI when item 11 is dropped; item 31 becomes clearly problematic and item 17 (solid non-bolded line) is less problematic but still cause for concern—likely becoming problematic if item 31 is also removed. Thus, while it seems like the other 31 items might provide enough information over the entire θ range, the removal of the problematic item leads to new problematic items in different ranges of θ than the original problem. Additionally, Figure 9b shows the IFTI values if the scale were only half as many items (i.e., only items 1–16 are shown). In this case, item 11 contributes almost 80% to the TIF. Unfortunately, we have whack-a-mole results: If the only problematic item is dropped, new problematic items are induced, and if the number of items is decreased, the effect of problematic items is heightened. This again suggests the poor identification of θ in these ranges, with most of the information being explained by one or two items in those ranges. The model is, therefore, too complex for the data supporting it.

## 4. Discussion

In this paper, we proposed a definition of a Heywood case in IRT, primarily focusing on the use of Fisher information. The definition we propose—the collapse of Fisher information, I(θ), to point masses—is principled and based on the IRT model. It can be made objective by choosing criterion values on the measures we proposed, for example, by choosing a particular value of I(θ) that is too high to be believed. Calculation can be performed by using numerical differentiation based on the predicted probabilities, so it is not necessary to perform difficult and error-prone analytic calculations. This helps make the detection of Heywood cases more automatic and objective than heuristic and is extremely helpful when dealing with models that have many and/or difficult-to-interpret parameters, e.g., the newer ones found in Asymmetric IRT or in polytomous models, among others.

Identifying Heywood cases is a very important consideration. While the effects can be mitigated with regards to inference about parameters, the effects on scoring are much more marked. Predicted latent trait levels and their standard errors, i.e., $\hat{\theta }$ and $\mathrm{SE}(\hat{\theta })$, are just as important as item-specific parameters are, and these quantities are strongly affected by the presence of a Heywood case, particularly when the number of items is moderate or small. The lack of identification signaled by a Heywood case has major implications with regard to scoring. When the information we have about θ as measured by Fisher information is weak, our ability to predict θ becomes more variable, as expected. While disappointing, the model provides its users with accurate knowledge. In the presence of a Heywood case, the exact opposite situation holds—an unwarrantedly large sense of precision that has to do with a failure of the model; thus, users should proceed with caution in such circumstances.

While the detection of Heywood cases is helpful, the next step involves avoiding them. The best approach involves designing measures and gathering datasets that are sufficient to support the intended analytic models. Of course, this is challenging, and psychometricians are frequently faced with analyzing the data they have, not the data they wish they had. We believe that recent work employing penalization on parameters via functions—such as ridge, LASSO, or via informative priors—is likely to be the best semi-automated approach. Regularization helps prevent these parameters from diverging to illogical values, though it does come at the cost of making misfit more likely. The basic approach was pioneered in the EFA literature by Martin and McDonald [48], who penalized the $\psi^{2}$ terms via priors. Penalization by quasi-Bayesian priors has been widely used since the 1980s in IRT for both estimation and scoring. Nevertheless, many models are still estimated by using unpenalized approaches, which are frequently software defaults. As such, poorly identified and/or overfit models appear in applied work quite frequently—see Rasmussen et al. [49] for a lengthy examination in the context of clinical psychology—but the problem is far from unique to clinical psychology. Fortunately, penalization has begun to be incorporated into popular software, e.g., Mplus [50], and hopefully will be incorporated into more software in the future. It is important to note, however, that the literature has not yet settled on how to make important choices about quantities such as priors or other tuning parameters of regularization. As we illustrated in Example 1, our measures may be useful for fine-tuning priors used to regularize IRT models, although the same data should not be used to both assign priors and evaluate them. Further research is needed in this area.

## Figures and Tables

**Figure 1 entropy-26-01096-f001:**
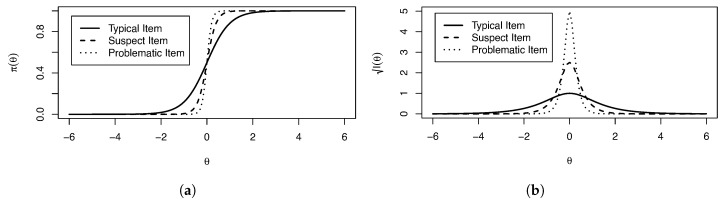
(**a**) IRFs and (**b**) IIFs for a typical, suspect, and problematic item. Note the IIFs are put on a $\sqrt{I(\theta )}$ scale for ease of visualization.

**Figure 2 entropy-26-01096-f002:**
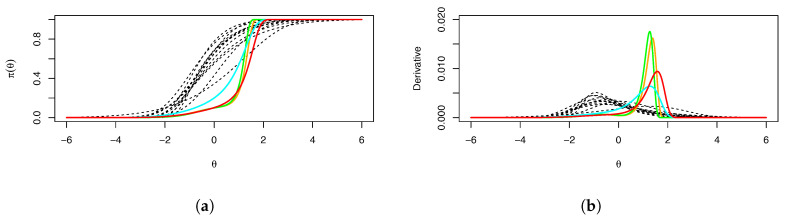
(**a**) Fitted π(θ) and (**b**) $\partial_{\theta }\pi \left(\theta \right)$ for Example 1. The mental rotation items are colored; item MR3 is orange, item MR4 is green, item MR6 is cyan, and item MR8 is red.

**Figure 3 entropy-26-01096-f003:**
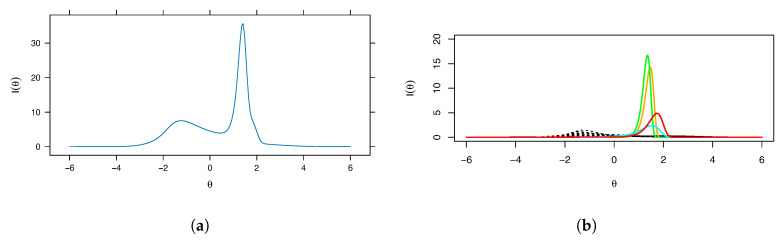
(**a**) Test information function and (**b**) fitted I(θ) for Example 1. The mental rotation items are colored; item MR3 is orange, item MR4 is green, item MR6 is cyan, and item MR8 is red. Note that the overall trend in the item information plots is reflected in the test information function.

**Figure 4 entropy-26-01096-f004:**
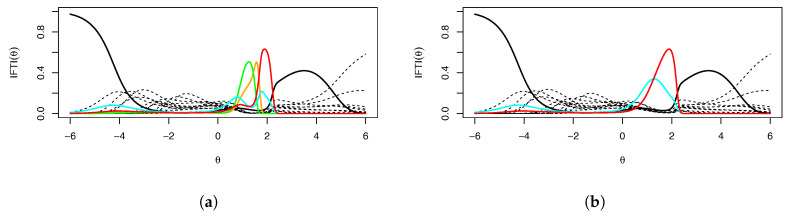
(**a**) Fitted IFTI for the total 16 items and (**b**) fitted IFTI for Example 1 with the two most severe Heywood case items removed. The mental rotation items are colored; item MR3 is orange, item MR4 is green, item MR6 is cyan, and item MR8 is red.

**Figure 5 entropy-26-01096-f005:**
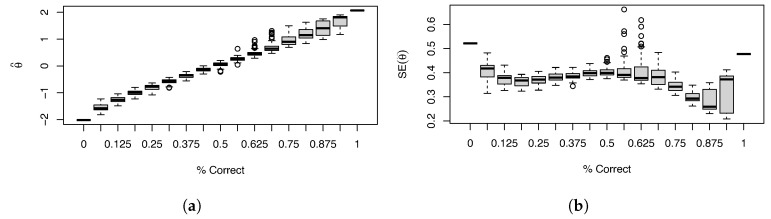
Box plots of (**a**) EAP predicted scores ($\hat{\theta }$) and (**b**) standard errors ($\mathrm{SE}(\hat{\theta })$), shown over the proportion correct. Note that while most boxes are fairly modest, for high proportions correct, the boxes are unexpectedly wide, indicating the instability induced by the mental rotation items.

**Figure 6 entropy-26-01096-f006:**
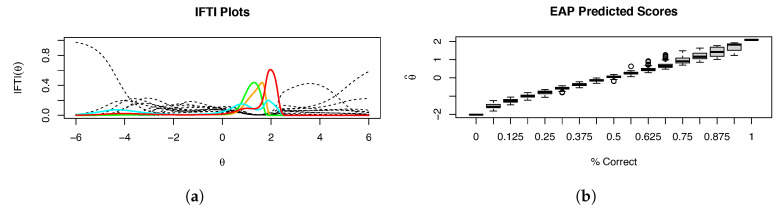
(**a**) Fitted IFTI and (**b**) box plots of EAP predicted scores for Example 1, with a loose N(0,1) prior set on the asymmetry parameter of the RH model. The mental rotation items are colored; item MR3 is orange, item MR4 is green, item MR6 is cyan, and item MR8 is red.

**Figure 7 entropy-26-01096-f007:**
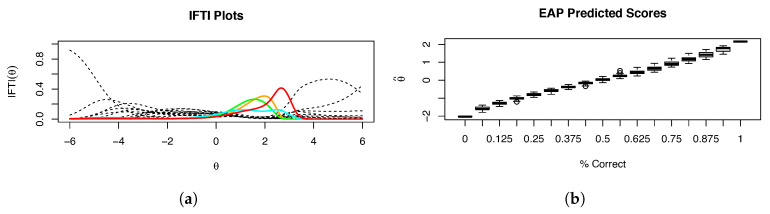
(**a**) Fitted IFTI and (**b**) box plots of EAP predicted scores for Example 1, with a strict N(0,0.25) prior set on the asymmetry parameter of the RH model. The mental rotation items are colored; item MR3 is orange, item MR4 is green, item MR6 is cyan, and item MR8 is red.

**Figure 8 entropy-26-01096-f008:**
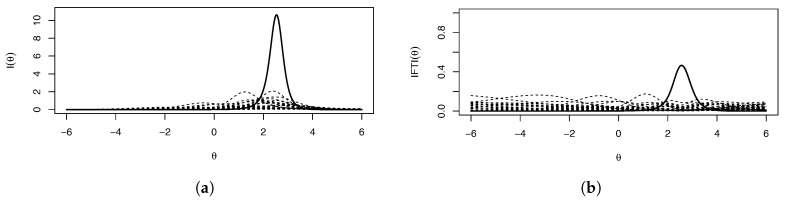
(**a**) Fitted I(θ) and (**b**) fitted IFTI(θ) for Example 2. Item 11 is bolded.

**Figure 9 entropy-26-01096-f009:**
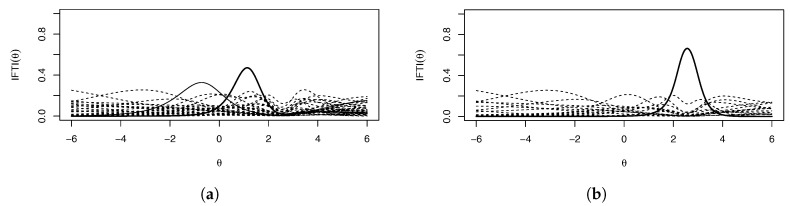
(**a**) Fitted IFTI(θ) without item 11 and (**b**) fitted IFTI(θ) without items 17–32 for Example 2. Items 31 and 11 are bold in **a** and **b**, respectively.

**Table 1 entropy-26-01096-t001:** Table of marginal easiness ($\mu_{j}$), Mokken $H_{j}$, coefficient estimates, and associated estimated bias and bootstrap standard errors for the RH model on the SAPA items. The M2 fit statistic for this model is 0.033.

				$\hat{\eta }$			$\hat{\xi }$			$\hat{\delta }$	
Item	${\hat{\mu }}_{j}$	$H_{j}$	Est.	Bias	SE	Est.	Bias	SE	Est.	Bias	SE
RE4	0.64	0.37	0.94	−0.003	0.066	0.47	−0.004	0.059	−0.12	−0.020	0.29
RE16	0.7	0.37	0.84	−0.003	0.056	0.69	0.002	0.057	0.28	0.003	0.32
RE17	0.7	0.41	1.02	−0.003	0.071	0.77	0.001	0.060	0.44	−0.002	0.28
RE19	0.61	0.34	0.82	−0.001	0.059	0.47	−0.002	0.051	0.94	−0.019	0.31
LT7	0.6	0.36	0.91	−0.002	0.056	0.39	−0.002	0.052	0.56	−0.003	0.27
LT33	0.57	0.33	0.79	−0.002	0.055	0.27	0.003	0.051	0.44	0.018	0.25
LT34	0.61	0.36	0.94	0.003	0.065	0.50	−0.001	0.055	1.05	0.005	0.37
LT58	0.44	0.38	0.84	−0.003	0.053	−0.18	−0.001	0.052	0.20	−0.006	0.24
MX45	0.53	0.3	0.62	−0.004	0.049	0.13	0.003	0.049	0.55	0.039	0.29
MX46	0.55	0.3	0.65	−0.002	0.045	0.17	0.003	0.046	0.26	0.010	0.21
MX47	0.61	0.33	0.78	−0.001	0.055	0.38	0.002	0.056	0.23	0.003	0.31
MX55	0.37	0.29	0.52	−0.003	0.041	−0.40	0.001	0.047	−0.29	0.003	0.28
MR3	0.19	0.47	1.03	0.003	0.066	−1.29	0.003	0.060	−2.45	−0.046	0.63
MR4	0.21	0.5	1.12	0.005	0.070	−1.29	0.007	0.059	−2.64	−0.112	0.76
MR6	0.3	0.43	0.91	0.001	0.060	−0.84	−0.002	0.052	−1.27	−0.128	0.51
MR8	0.19	0.45	0.94	0.006	0.059	−1.27	−0.004	0.060	−1.53	−0.079	0.43

## Data Availability

The original contributions presented in this study are included in the article/Appendix A. Further inquiries can be directed to the corresponding author.

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
