# Peer review of "A Definition of a Heywood Case in Item Response Theory Based on Fisher Information"

_entropy, 2024, doi:10.3390/e26121096_

Round 1
Reviewer 1 Report
Comments and Suggestions for Authors
Page 1 line 3
The authors have submitted their manuscript to the journal Entropy. But there appears to be no mention of entropy in the manuscript. A motivation is therefore needed. (In fact, it is well known that entropy plays a major role in this field, as exemplified by the references given below.)
Line 25
What is a "fixed form"?
Line 28
...six "fill in the blank" lines...
Lines 60 - 61
Is that "test persons" and "items" - or something else?
Line 69, eq. 2
I would like to see a bracket around the sum (lamdba + Psi), since I assume you mean a Normal distribution with mean u and variance equal to that sum?
What is Λ? Is it a matrix of the "factor loadings", λi, since you specify (at lines 61 - 62) your code upper-/lowercase ?
Line 76
Sounds like Lambda is a "design matrix"? (not a matrix of factor loadings - see query above, at line 69).
Line 81
Why is it "inconsistent" to be a "covariance matrix"?
Line 86
Are "Heywood cases" unique to GLMM or are there other kinds of regression where they might be found?
Line 88
Is this a result of the well-known central limit theory applied to a multivariate case? Please also clarify whether the results are different if the multivariables are uncorrelated or not - i.e. the difference between multivariable and multivariate situation.
Line 91
Do you mean in the present manuscript or somewhere else in the literature? Please give a reference.
Line 93
What do you mean by a "measurement model"?
Line 97
See my query above (line 76)!
Lines 98 - 9
Infinite weight can be given for other reasons - for example where a metrological reference anchors the regression, but such cases are not "ill-defined". Please clarify.
Line 122
Please give a reference - e.g. the "counted fractions" of J.A. Tukey, Chapter 8, Data analysis and behavioural science, in The Collected Works of John A Tukey, Volume III, Philosophy and Principles of Data Analysis: 1949 – 1964, ed. by L. V. Jones, (University North Carolina, Chapel Hill, 1986)
Line 126
Isn't delta also an IRT latent variable?
Line 127
Unnecessary repetition!
Line 129
Do the authors mean Spearman's 'two-factor theory of intelligence ': "Charles Spearman developed his two-factor theory of intelligence using factor analysis. His research not only led him to develop the concept of the g factor of general intelligence, but also the s factor of specific intellectual abilities." [Wiki]
Line 130
Why not refer to the Rasch (1960) IRT model instead? His principle of specific objectivity is to my knowledge the only GLM with which separate estimates of person ability and item difficulty (sic!) can be made reliably. Please clarify.
(There are plenty of modern resources aout there for Rasch analysis: WINSTEPS, RUMM, erm and other R, etc. Rasch, G. (1960). Probabilistic models for some intelligence and attainment tests. Copenhagen, Denmark: Danmarks Paedogogiske Institut. (reprint 1980, Chicago: University of Chicago Press))
Line 130
The item slope is normally called the "discrimination", whereas "loading" is common terminology in PCA - the authors' terminology is confusing (unless they can convince me that these different concepts are connected).
Line 140
These are different kinds of concept: Bernoulli and Poisson are classic statistical distributions while categorical often refers to classification (on for instance a nominal or ordinal scale). Please clarify.
Line 164
Not exclusively - the logistic Rasch model is often applied also to polytomous (multinomial) cases.
Line 168 - 9
Don't the authors want to admit the Rasch model into their arguments, which is perhaps the mostly widely used logistic model - see line 130? What's the issue which explains the authors' reticence?
Line 176
Without a reference to what the authors mean by "graphing" (casual or not!), it is difficult to defend the method against the authors' criticism. The IRT literature abounds with accounts of graphing - such as PCA loading plots and construct alleys - see Rasch literature (ref. line 130) and as exemplified in Melin, et al. 2022, The Role of Entropy in Construct Specification Equations (CSE) to Improve the Validity of Memory Tests: Extension to Word Lists. Entropy 24, 934. https://doi.org/10.3390/e24070934
Line 178
The manuscript could - in my opinion - start here instead - I suspect you are going to lose a lot of readers who find the "elegant" material to this point rather heavy-going. (I can understand that the authors want to show off their general theory, but I'm sorry I don't see the added value.)
Line 195
The authors should refer to earlier work which has studied how the first derivative of the IR can reveal deviations from the basic model - see for example, Appendix C of Melin, et al. 2022 https://doi.org/10.3390/e24070934.
Line 198
The partial derivative of eq. (10) is given for example by: Pendrill 2019 eq. 5.25. Your expression looks different: please clarify.
(Pendrill 2019, Quality Assured Measurement – Unification across Social and Physical Sciences, Springer Series in Measurement Science and Technology, ISBN: 978-3-030-28695-8 (e-book), https://doi.org/10.1007/978-3-030-28695-8)
Footnote to page 5
In calculus, one has derivatives (as here) as opposed to integrals. In econometrics a "derivative" is a financial tool to make gains on changing equity prices. Please clarify.
Reviewer 2 Report
Comments and Suggestions for Authors
Thank you for the opportunity to review this paper. From my perspective, a significant contribution of this paper is providing language and rationale for flagging items with too-high-discriminations as problematic. Although many IRT practitioners are familiar with this problem, naive users may not be aware of this issue, and the authors correctly point out that there are currently no standards for flagging such items in IRT.
Overall, I find the content of this paper to be correct and useful, but it could benefit from a more careful edit. In some places, the text is not specific enough or otherwise somewhat inaccurate; examples:
- p. 3, line 111 (4th item on the list). Sampling variability is not a model specification error, like other items on the list are. Perhaps the list is better described as conditions that could lead to Heywood cases.
- p. 5, line 201 "we tentatively adopt this as an indication of a likely Heywood case in analogy to a too large regression weight in logistic regression, wherein a unit change on the regression scale would lead to a predicted probability change of nearly .5 [20]". Of course, the impact on predicted probability of a unit change in the logistic regression coefficient depends on the starting value (which is expressed more precisely in reference [20]).
- p. 6, line 227 "I(theta) is advantageous over [IRF first derivative], as it exists for all models" - I don't understand how you can calculate information if the first derivative doesn't exist...
In other places, the text would benefit from including less information. Examples:
- There is perhaps more notation and abbreviations than needed, making the paper more difficult to read. The authors may consider omitting some of the abbreviations or notation or at least providing a table describing the referent of each abbreviation and symbol.
- Because this paper is focused on IRT, I think that it is fine to assume a standardized theta throughout the IRT sections. The only routinely used IRT models that don't assume a standardized latent variable are from the Rasch family, which shouldn't have issues with Heywood cases due to constant discriminations.
- It's not clear to me how equation 13 advances the developments in this study. IT seems to be a general comment on the information function (?)
The IFTI is a potentially useful metric, but might it be less useful if there are multiple Heywood cases with roughly the same location? And if a Heywood case occurs at extreme theta, might WIFTI not flag it? I would like to see a discussion of these potential implications.
Also, it's unclear whether you propose calculating IFTI and WIFTI for only theta* or at multiple theta values.
The effects on scoring in Figure 5 could be more compelling if compared to an example without Heywood items. As is, it's not clear to me how significant the effect is.
p. 12, line 395: "The best approaches involve gathering datasets that are sufficient to support the intended analytic models and making use of properly specified models insofar as possible" Arguably, for many applications, the best approach is designing *measures* that don't lead to these extreme estimates.
The authors' idea to develop an IRT program that automatically checks for Heywood cases is a good one and would be a useful contribution. I had hoped to see a concrete suggestion for what this add-on program could look like, perhaps in the form of a wrapper function that could be used with the mirt package. I anticipate that the ideas in this paper could see much greater dissemination and application if such a function is included as supplementary material.
Reviewer 3 Report
Comments and Suggestions for Authors
The manuscript introduces a definition and analysis of Heywood cases within item response theory (IRT) models and offers novel ideas for examining functionals of item response functions (IRF). However, I do not agree with connecting this study to Heywood cases as understood in confirmatory factor analysis (CFA).
General Comments:
1. The arrow notation for vectors is unnecessary and outdated; lowercase bold font is sufficient.
Major Comments:
2. Line 6: It is unclear why Guttman items are necessarily problematic. If the IRT model holds, these items are actually highly discriminative with respect to \theta.
3. Following on from Comment 2, would the authors recommend removing items with high discriminations (or small residual variances) from IRT/CFA models? Such items are generally considered "reliable," so discarding them seems contrary to psychometric principles. This is not necessarily an issue, but the arguments presented are unconvincing.
4. I interpret the authors' concerns as primarily related to IRT scoring methods, where they seem reluctant to allow certain items a large influence. This relates to the work of Camilli (2018) and Chiu and Camilli (2013).
5. Line 34: Boundary solutions are more an issue in statistical inference than in point estimation.
6. Line 105: Wouldn’t constrained or regularized estimation address Heywood cases? Such adjustments might be especially relevant in small samples, more so than model simplifications (point 4, line 111).
7. Line 315: Why are EAPs considered the “industry standard”? They introduce significant shrinkage, and WLEs are arguably more common.
8. It may be worth removing all references to Heywood cases and reframing the paper around IRT scoring. Improper solutions seem more connected to the global IRT model rather than locally defined item response functions.
Minor Comments:
9. Line 73: “Positive semi-definite” or “positive definite”?
10. Line 80: Why are the estimates called “inconsistent”? This seems inconsistent with the statistical definition of consistency.
11. Throughout: Should it be “Thomson” instead of “Thomsen”?
12. Line 122: Should "binary link function" be emphasized over "link function"?
13. Line 165: Briefly describe the desirable properties of the logit link function.
14. Line 178 and elsewhere: Distinguishing \Theta from \theta seems unnecessary; I suggest using \theta throughout.
15. Equation (14): A parenthesis is missing.
16. Line 227: Is I(\theta) significantly more complex to compute than \partial \pi (\theta)?
17. Equation (15): Use a different index than jjj in the denominator sum.
18. Equation (16): In its current form, f(\theta) cancels out, equating WIFTI and IFTI. Did you mean integration over \theta?
19. Line 387: It is unclear how LASSO or sparse regularization methods would help here; ridge-like (dense) regularization might be more appropriate.
Camilli, G., 2018. IRT scoring and test blueprint fidelity. Applied Psychological Measurement, 42(5), 393-400.
Chiu, T.-W., & Camilli, G. (2013). Comment on 3PL IRT adjustment for guessing. Applied Psychological
Measurement, 37, 58-68.
Round 2
Reviewer 1 Report
Comments and Suggestions for Authors
Thank you, authors, for providing us with a much improved manuscript.
- There is a long history of debate between IRT-supporters and the Rasch school which will surely continue. In the revised manuscript there is now a fairer chance for readers to consider the authors' approach in relation to what is possible with the Rasch model (which cannot be ignored only on the grounds of being judged too "simplistic").
- It is never too late to check your understanding of the elementary (and I don't mean simplistic) concepts. Sure, references [2] and [3] are often quoted, but those works missed the elementary insight of Tukey. His famous (but now largely forgotten) account of counted fractions is a necessary precursor to any analysis of asymmetric ICCs.
- The authors have provided me as referee some generous texts. I'm flattered, but the important thing is however that their insight be shared with the readers of this publication.
Reviewer 2 Report
Comments and Suggestions for Authors
I am satisfied with the authors' revisions, especially in the context of the limited editing and review time allowed by the journal.
Reviewer 3 Report
Comments and Suggestions for Authors
I comment on some remaining points:
1. The authors still like to use their vector arrow notation. They should consistently use their preferred style if they want to insist on this. For example, Eq. (3) and (4) also require arrow notation for Y.
2. My Comment 17 in the previous review still applies. A different summation index is required in Equation (14).
3. Sect. 2.2.3: To avoid thresholds for IFTI dependent on J, I would rather multiply your IFTI by J. Then, the item weights are compared to a relative size of one.
4. I think item weights regarding test information, as presented to you, should be related to local item scoring weights used in likelihood inference for persons. That is, it is unclear why the interpretational focus should be on item contributions in the information function instead of item impact in the local scoring, which has direct interpretations to the \theta variable.
5. I still do not see arguments in the manuscript as to why Guttman items are problematic.
Round 3
Reviewer 3 Report
Comments and Suggestions for Authors
---